# Perceptions, Attitudes, Experiences and Opinions of Tuberculosis Associated Stigma: A Qualitative Study of the Perspectives among the Bolgatanga Municipality People of Ghana

**DOI:** 10.3390/ijerph192214998

**Published:** 2022-11-14

**Authors:** K. A. T. M. Ehsanul Huq, Michiko Moriyama, David Krause, Habiba Shirin, John Koku Awoonor-Willaims, Mahfuzur Rahman, Md Moshiur Rahman

**Affiliations:** 1Graduate School of Biomedical and Health Sciences, Hiroshima University, Hiroshima 734-8553, Japan; 2Clinical Research Center, University of Massachusetts Medical School, Worcester, MA 01655, USA; 3Public Health Consultant and Health Systems and Policy Analyst, Accra M 44, Ghana; 4International Centre for Diarrhoeal Disease Research, Dhaka 1212, Bangladesh

**Keywords:** tuberculosis, stigma, Bolgatanga, Ghana

## Abstract

Tuberculosis (TB) is the tenth leading cause of death worldwide. About 1.3 million people die from TB each year, and most of them are in developing countries. The stigma associated with TB is a barrier to seeking treatment and adequate care. It causes a delay in treatment-seeking and diagnosis and thus decreases the likelihood of being cured and ultimately leads to death. The objective of this study was to explore the perceptions, attitudes, experiences, and opinions about stigma related to TB among adults infected with TB and adults who were not infected with TB. Our study was qualitative in nature. The study was conducted in the community of Bolgatanga municipality of the upper-east region of Ghana. Three focus group discussions (FGDs) were conducted; one with six TB-infected females, one with seven TB-infected males, and one with six non-TB-infected participants. Data were analyzed using qualitative content analysis and presented in pre-defined and/or emerging themes: perception about signs and symptoms observed by TB infected person, attitudes towards TB patients before and after diagnosis, reasons for stigmatization, perception about diagnostic testing, and taking the drugs. Transcripts of the discussions were read, and a list of meanings for units, codes, and themes was generated on the research question. We identified the existence of stigma associated with TB. TB-infected male patients had more autonomy in decision-making about receiving treatment and other family matters compared to female TB patients. TB-infected women suffered more economic vulnerability due to the loss of their work, and the stigma was worsened due to delayed diagnosis and treatment, and they were regarded as liabilities rather than assets. TB-infected patients became stigmatized because non-TB-infected community participants did not want to come into close contact with them. Our findings suggest heightening of advocacy, communication, social mobilization, and health education on TB in the community of Bolgatanga municipality is needed to allay TB-related stigma, especially for women.

## 1. Introduction

### 1.1. Background

Globally, tuberculosis is one of the top ten causes of death, and 90% of cases occur among adults aged ≥ 15 years. In 2020, it was estimated that about 1.3 million people died compared to 1.2 million in 2019. Moreover, due to the almost halt of declining TB incidence, the forecast is to be worse in 2022. It is still not fast enough to achieve an end to the global TB epidemic, which is a 75% reduction in TB deaths by 2025. Furthermore, drug-resistance TB is becoming a continuous public health crisis [1]. TB is considered a dirty disease, and the patients feel shame. They isolate themselves by withdrawing themselves from social contact and other people. With TB, they fear the consequence of many difficulties in their individual, family, and social life [2]. Fear of stigmatization makes TB patients more vulnerable because many choose not to disclose their disease and abstain from seeking treatment [3]. People are also afraid of TB infection, which makes it stigmatized. Women suffer more malicious socioeconomic consequences with TB stigma. It causes a delay in the diagnosis and noncompliance with receiving treatment [4]. Misconception about TB transmission and negative attitudes toward TB patients is related to stigmatization [5]. Even TB stigma persists after the complete recovery of this disease. Therefore, only about half of the patients feel mentally happy after the completion of TB treatment [6].

Drug-resistant TB-infected patients experienced more stigma due to their incurable and infectious state [7]. TB treatment delay increases morbidity, mortality [8,9], and the spreading of this contiguous disease among close contact and communities [8,10]. Ghana has adopted a directly observed treatment short-course (DOTS) strategy by implementing a National Tuberculosis Control Programme (NTP) since 1994 with full integration into the national health system and achieved 100% DOTS coverage in 2005 [11]. The Ghana National TB Voice Network empowers TB-infected patients and community participation through effective advocacy, communication, and social mobilization to reduce stigma and increase treatment adherence [12]. Since 2009, according to the World Health Organization global lists, Ghana has been considered one of the forty-one high-burden countries for TB/Human Immunodeficiency Virus (HIV) globally [13]. The situation became worse in 2016–2020, when it was ranked among the thirty high TB/HIV burden countries and considered endemic (100–299 new and relapse cases per 100,000 population per year) in 2019 [14]. TB patients in Ghana, including their households, are facing enormous catastrophic costs, amounting to about 75% of the total household cost in 2016–2020 [1]. 

In Ghana, people living in overcrowded and poorly ventilated rooms, patients with HIV/AIDS, diabetes, cancers, kidney failures, malnutrition, alcoholics, and smokers are the most susceptible to getting TB. NTP-adapted TB preventive therapy (TPT) is to save those in high-risk populations [15]. People with a high burden of TB coupled with HIV and AIDS double the suffering of the patients [14]. The last nationwide population-based prevalence survey was conducted in 2013 and revealed a much higher TB burden compared to the previous estimation. The main concerning findings were most of the TB cases were smear-negative with abnormal radiography and without having chronic cough [16]. Another drug resistance TB survey was conducted in 2016–2017 for the first time nationally and found higher levels of multi-drug resistance to TB and overall resistance to any TB drug among the previously treated TB patients [17]. Despite enormous efforts countrywide to develop capacity, stigmatization remains a real obstruction to eradicating TB by 2030. Furthermore, drug-resistant TB increased from 30 to 198 cases between 2013 and 2017. It is also concerning that TB case notifications declined from 15,606 to 14,550 between 2013 and 2017 [18]. The National tuberculosis health sector strategic plan 2015–2020 included a comprehensive communication strategy to reduce TB stigma as its main intervention. The plan focused on knowledge, the stigma of TB, and healthcare workers (HCWs) service-related barriers to TB stigma [19].

### 1.2. Purpose

We conducted this study to explore the perceptions, attitudes, experiences, and opinions about stigma related to TB among adults infected with TB and adults who were not infected with TB.

### 1.3. Significance

This is the first study to explore the perceptions, attitudes, experiences, and opinions of TB patients and community members among the people of Ghana. By exploring the factors related to TB stigma and addressing them, we can manage more TB patients through increasing awareness, and thus, we can reduce TB-related burdens in the communities.

## 2. Materials and Methods

### 2.1. Study Population, Data Collection, and Study Design

The study was conducted at Bolgatanga municipality in the upper-east region of Ghana between November 2012 and January 2013. There has not been much changed over the last 10 years of TB situation regarding TB stigma fears or knowledge in the community. It has a total population of 139,864, with 47.6% males and 52.4% females in 2021 [20]. The participants were from different ethnic backgrounds, including the main ethnic groups in the region, the Grunis, Akans, Ewes, and Ga-Adanbge. There are other distinct minority ethnic groups, such as Moshie, Mamprusi, and some others who migrated from neighboring countries, such as Burkina Faso, for the purpose of farming and petty trade [21]. In Ghana, adult literacy rate was 71.5% in 2010, and it increased to about 79% in 2018. However, in Bolgatanga Municipality, it was 64.6% in 2010. Among non-literacy, females and males were 42.6% and 27.2%, respectively [21].

#### 2.1.1. Groups of Participants

The study was qualitative in design. Three face-to-face focus group discussions (FGDs) were conducted; one with six TB-infected females, one with seven TB-infected males, and one with six non-TB-infected participants with both sexes. At first, we started mixing FGDs with TB patients and realized that men were more dominant in discussion, and that made women not participate fully. A decision was therefore taken to separate men from women in order to make each group feel free and relaxed to express their views. A total of ten participants refused to participate due to mainly fear of being discriminated against for TB in this study. On average, each FGD took about sixty minutes. TB and non-TB participants were purposively selected from the Bolgatanga Health Clinic and from the community, respectively. TB patients were identified from the list of clinic registers provided by the consultant of Bolgatanga Health Clinic, and non-TB participants were identified by the community leaders. FGDs for the TB participants were held in one of the consulting rooms in the TB clinic to maintain privacy. The FGD for the non-TB participants was identified by the local community leaders and was held in a small room (hall) in one of the community leader’s houses. The landlord or any elderly non-TB person in the household was selected for discussion. Local community leaders mobilize and guide the community members and facilitate the problem-solving and decision-making process for the benefit of the community itself. They facilitated the FGD; however, they did not participate in the discussion. As the participants resided in the same community as the community leaders’ house, this gave opportunity to participants to feel comfortable about their opinions and attitudes how they felt about TB patients. 

#### 2.1.2. The Inclusion and Exclusion Criteria of the Participants

We enrolled TB-infected patients and non-TB participants who were living in the study area, aged 18 years and above, both male and female, and were willing to participate in the study. Every second house without windows was selected for non-TB participants. This was because it was thought that overcrowded and less ventilated rooms were suitable places for spreading TB to close contacts. 

#### 2.1.3. Study Participants Recruitment Procedure

The ‘Discussion Guideline’ (Table 1) and the data collection method were discussed with the guidance of a female qualitative researcher who was the Professor and acted as a supervisor of this research before embarking on the data collection. The FGDs were moderated by a male principal investigator who was a master’s student (D.K.) and had experience and training in qualitative research. He was assisted by one assistant moderator who had vast knowledge of TB research. Assistant moderator operated as a translator. The role of translator was considered important since the investigator did not know the local language. This person was a community health worker who had been working on the issue of TB with immense knowledge and experience. She knew almost all the local languages of the study areas. For easy understanding of the research, at first, the discussion was written in the local language called Frafra and further translated into English. The purpose of the study was well explained to all the study participants. Participants were all made aware that they had the choice to withdraw from the study at any point in time, and no one would be penalized for refusing to participate. Before conducting the FGDs, rapport was built with the participants.

#### 2.1.4. Transcription and Translation Procedures 

The FGDs were audio recorded. Notes were also taken to capture emotional atmosphere, gestures, and signs made by all participants for cross-referencing. A pre-test discussion was carried out among five participants from the same community to test the feasibility of using data collection guidelines and check the discussion topics or concepts if they were comprehensible to the study participants. Those five test participants were excluded from further participation in the main FGDs. Pre-testing helped us determine if participants understood the questions as well as if they could perform the tasks or have the information that questions required. The guide was sufficiently explicit, objective, and comprehensive and presented or not questions that could be ambiguous or equivocal. First, we started with the FGDs by introducing different pictures of a TB client in the TB clinic; this gave us access to the opinions and attitudes of all the participants. The FGDs started with a broad question like: “Who did you first talk to about your TB condition”? The response was followed by a probing question. Questions were modified as further insights emerged during data collection (according to the Lincoln and Guba theory of emergent design) [22]. Additionally, a TB poster was introduced halfway through the FGDs to stimulate participant responses. This stimulated reactions among participants, which were clearly seen, making them elaborate on points made by others. These interactions with different people in the discussion specifically brought out various opinions and great experiences that might not emerge during semi-structured interviews. As a result, information was forthcoming during the discussion as we continued to ask probing questions.

#### 2.1.5. Trustworthiness

Trustworthiness is referred to by Nowell, Norris, White, and Moules as credibility, transferability, dependability, and confirmability [23]. The validation was performed through the oversight of a supervisor who had a doctoral degree with a vast experience in qualitative research. This involved reviewing the ‘Question Guide’ and listening to the tapes at the close of the day’s work, and all doubts were resolved in the presence of the principal investigator and translator. The audio tape recordings were transcribed from Frafra language word by word into English. When the first data were collected, the informants and the investigator listened carefully to the tape recordings to be sure of understanding. All the recordings during the discussions were cross-checked carefully, which gave a deeper understanding of the information and answers. During the translation, the translator worked in a very quiet environment to be able to listen and refer to the recordings in case of any doubt. There was no disturbance or interruption during this process. Notes were taken during gestures; simultaneously, audio recording was made, and signs were considered along with the comments recorded. All data were revised twice, which helped the researcher capture the non-verbal expressions of the clients during the discussions. Field notes were referred to and compared several times to capture the emotional atmosphere of participants and to clarify what they really meant. 

#### 2.1.6. Human Subject Protection

Investigators who were directly involved in recruitment process of the study participants took the National Institute of Health (NIH) protecting human research participants course before initiation of the study.

#### 2.1.7. Operational Definitions of the Concepts

Stigma: Stigma described as the processes of different labeling of individuals, stereotyping in negative way, cognitive separation, negative emotional reactions, status loss, and discrimination that lead to detrimental consequences [24].Perception: Perception is the process of individual feeling, experiences, needs, motivation, and educational background and concludes with thinking, analyzing, and interpretation of the environment [25].Attitudes: Attitudes describe a person’s behavior towards other persons or objects. Attitudes towards different individuals, institutions, and social issues reflect the way to perceive the world around the person [26].Opinions: Opinions are evaluative beliefs and are usually narrow in content or scope. They involve individual’s judgments about the probability of events or associations regarding some object on specific dimensions [26].

### 2.2. Data Analyses

The data were analyzed by qualitative content analysis (QCA). No models or theoretical frameworks were used in this analysis. Emergent design was applied to gain further insights during the FGDs [27]. We performed FGDs rather than individual in-depth interviews, as we considered FGDs to be the best way to exchange viewpoints and attained the discussion of disagreements among the participants even though people discussed sensitive issues sensitively due to stigma.

We did ongoing analysis during the data collection process. After completion of three FDGs, we had a robust and valid understanding of the study phenomenon with same findings that ceased to provide new information. We felt that three FGDs were sufficient to achieve the study objectives and reached the saturation point. 

First, the authors thoroughly read and reviewed the focus group transcripts to identify meaning units, and some of the parts that relate to each other were combined. A codebook was established after reading through the focus group transcripts. It was modified based on continuous discussions among the authors (data analysis team included three members). The meaning units were then condensed, from which the initial set of codes was generated. The codes were finalized once the authors reached a consensus. The initial codes were applied to a small sub-set of focus group transcripts, which set the basis for the generation of themes. Then, based on that, the key themes were identified and then incorporated into the thematic matrix. Based on this, the relevant data were categorized into the matrix accordingly (Table 2).

Furthermore, inter-rater reliability was established by using the collaborative coding framework where the authors sit together and discuss their feedback about the assignment of each theme. All of these decisions were documented through decision trials. If there were any discrepancies, these were discussed and then solved by the senior team members.

We used qualitative software (Nvivo, version 9) for our data analysis. Data were analyzed with four main themes: (1) perception about signs and symptoms observed by TB infected person; (2) attitudes towards TB patients before and after diagnosis as being infected; (3) social consequences of stigma towards TB, and (4) perception about taking the drugs. Finally, we reported our study findings using consolidated criteria for Reporting Qualitative research (COREQ) checklist [28].

## 3. Results

### 3.1. Socio-Demographic Characteristics of the Participants

At the beginning of the focus group, the researchers collected their socio-demographic information. This was not a separate survey or questionnaire; rather, it was embedded within the focus group guideline. Three focus groups were carried out, and a total of 19 participants, 10 males and 9 females between the ages of 27 and 55 years old. Five participants had the highest educational level of junior secondary school. Seven participants had some form of education, at least up to primary-three levels and another seven participants had no education. Most of the respondents (eleven participants) were farmers, and the remaining eight participants were traders. Regarding religion, nine of the participants were Traditionalists; they traditionally expressed their belief in a supreme God as Nyame or Mawu [29], five were Christians, and five were Muslims (Table 3).

### 3.2. Data Are Presented with the Following Four Themes

#### 3.2.1. Theme #1: Perception of Signs and Symptoms Described by TB Infected Patients

From the discussions, the participants said that before starting treatment, they vomited soon after taking in food, experienced pains in the ribs and chest, and had difficulty in breathing. When these signs were experienced, they were taken to a hospital where diagnosis and chest radiographies were performed to confirm TB infection.


*“Before we came to the hospital, I vomited when I ate. I also complained of rib and chest pains. So, it was my son who even told me that I was sick and when we came to the hospital, they took radiography and then told me it was TB.”*
—Female TB patient #1

Another respondent, fifty years of age and with nine years of schooling, said:
*“I had pain on my chest and when I cough it contracted. I knew it was TB and when I told people about it, they said that was TB. So, I must come to the hospital because the way I was coughing if I stay at home with it would not be fine. So, I came here with my brother who is a teacher, and the doctor said it was TB.”*
—Male TB patient #1

It also came to light that some of the patients had persistent sickness, which was treated several times by some other health facilities before they were finally diagnosed as having TB at the hospital- as indicated by another participant in the FGDs.


*“When it started it didn’t keep long, and I took medicine, and it stopped. It came again and I took medicine again, and it stopped for the third time. I didn’t know what was wrong with me, so I came to the hospital, and they told me it was TB.”*
—Male TB patient #2

When asked why they decided to resort to treatment from the hospital and no other place, one participant of thirty-eight years old and six years of schooling completed said:


*“I know I cannot get treatment elsewhere that’s why I came to the hospital to get help.”*
—Female TB patient #2

However, some of the TB patients in this study said that they were advised by relatives and family members to go to the hospital for a checkup since they did not know what was wrong with them. One patient of twenty-eight years old who had eight years of schooling completed a said during the discussion:


*“There are people who were already aware that I was not well. The way I was complaining of my ribs and chest pain and vomiting after eating, they all knew that there was something wrong with me and advised to come to the hospital.”*
—Female TB patient #3

Patients admitted to the infectious disease ward were requested to take chest radiography, and positive patients were informed that they were infected with TB.

Source of Infection

The TB patients said that they did not have any idea or know exactly where they might have contracted the infection. One of the females, however, was suspected of getting it from eating with her family members, though she also made it clear that no one in her family had the infection before. 

Continued by another patient, twenty-nine years old and with six years of schooling, completed


*“I think when I eat with my sister, and she bites the meat in her mouth, and I eat the rest, because she smokes tobacco and cough.”*
—Female TB patient #5

One male TB patient also said he had been sleeping outside. 

One participant, forty years old and with five years of schooling, said:


*“I slept outside and at night, I felt cold. You know cough comes with cold so as I slept outside, I got It.”*
—Male TB patient #3

Some patients expressed their belief that no one was to blame for their infection since it was a sickness that came on its own.

A patient of thirty-five years old and eight years of schooling completed expressed this with the following:


*“I will not blame anybody because it is a sickness that has come, and it is not somebody who put the sickness in me.”*
—Male TB patient #4

A TB patient of thirty-nine years old and ten years of schooling completed also said:


*“Nobody gave me the infection. I don’t know where the infection came from, I can’t tell.”*
—Female TB patient #4

Duration Before Seeking Hospital Treatment

The participants said that they sought treatment from the hospital after they had a persistent cough for one month or more. Only one respondent visited the hospital after a shorter period of eleven days of coughing. 

This patient of forty-one years old and six years of schooling completed said in the FGD:


*“I didn’t know that it was serious but within the 11 days, I was having chest pain. My wife said I should come to the hospital because I was also smoking.”*
—Male TB patient #5

Another patient, twenty-nine years old and with seven years of schooling, completed the FGD: 


*“… I had a cough for more than three weeks and felt pain; I stopped smoking tobacco but still felt pain, so my brother said to come to the clinic.”*
—Male TB patient #6

A participant said:


*“I had chest pain for two weeks, when I cough the right side of my chest was paining. I took tablet paracetamol, but still felt pain. So, I told my son, and we came to the clinic.”*
—Female TB patient #5

Another patient of thirty-one years old and eight years of schooling completed continued:


*“… first, I felt sick for some time and it stopped, and it came again, by then I went to the southern part of the country to attend a relative’s funeral home at Bolgatanga. When I returned at home it attacked me, and my family brought me to the hospital.”*
—Female TB patient #6

The TB patient was aware of the mode of transmission of TB infection among close relatives, who, in most cases, could be either the wife, the father, or the son/daughter. They might be infected either by staying together at home or being present at the time of diagnosis, as expressed by a male participant, fifty-one years old and with no education:


*“As for me, I had cough for ten days, I told my wife and my father…we were staying together, and they are my relatives.”*
—Male TB patient #7

#### 3.2.2. Theme #2: Attitude towards the TB Patients before and after Diagnosis

Attitude of Relatives

Divergent views were expressed when the participants were asked about the attitude of their relatives after they were diagnosed as having TB. Participants admitted that the attitude of their relatives had not changed towards them. However, in contrast, relatives of other patients neglected them after TB was diagnosed. The view was expressed as the following quote indicates:


*“When they told me I had TB, my son went back and told our people, and they came only once and never came again. It is now only my son who has been coming.”*
—Female TB patient #1

Another patient continued:


*“My husband told my sister that she had a long-lasting cough (korsi-kheka), and she was in the hospital; she never came to visit me.”*
—Female TB patient #2

One respondent said about his wife’s attitude before and after he was diagnosed as having TB:


*“No, her attitude has not changed.”*
—Male TB patient #2

Another patient continued:


*“As for me, my families don’t shun me; my son and my wife come here and visit me.”*
—Male TB patient #3

Attitude of Friends

Friends’ views were not very different from those of the relatives. It was said, however, that females were more neglected as most of the males said their friends visited them. The FGD with the female TB patients suggested that their friends did not care much about them but just abandoned them.

A TB patient expressed these words in the local dialect: 


*“They neglect us, nobody comes, they know I have a long-lasting cough.”*
—Female TB patient #3

One of the participants even said, *“I don’t have a friend”* when she was asked about the attitudes of her friends towards her after they got to know she was infected with TB—Female TB patient #4

Another patient said:


*“After bringing me here to the hospital by my sister, none of my friends come to visit or ask my sister how I am feeling, they know l had korseh kheka (long-lasting cough).”*
—Female TB patient #5

The story was different in the FGDs with the male patients: *“Yes they have been coming”*—Male TB patient #4 when asked whether his friends still visit him.

Similarly, another TB patient said:


*“My friends come to visit me after I have been in the hospital for one week and brought some oranges.”*
—Male TB patient #4

Attitude of Community Members

Though it was clear in the FGD with the male TB patients that some community members still visit the TB patients, the FGD with the female TB patients indicated otherwise.


*“It is normal they come to visit me because they are not the people who gave me the sickness.”*
—Male TB patient #5

Continued another patient:


*“People come to visit me every Saturday and I play card games and they told me not to be worried because in the hospital l would get better.”*
—Male TB Patient #6

A female patient said:


*“(Respondent took a deep breath) … As I am here, some of them did not come to visit me. At first, when I was at home, some would not even ask me how I was feeling.”*
—Female TB patient #6

The FGD, with the female non-TB participants, talked about the plight of female TB patients. This shows a sign of stigmatization towards TB patients. 


*“I will not get closer to that person because I know TB is something that is contagious and dangerous so if I get close to the person, he/she will infect us with the sickness he/she has gotten.”*
—Non-TB participant #2

Community Perception of TB Patients

When the non-TB participants were asked about the community’s perception of TB-infected people, they said that they were also human like them (the community members). The only difference, however, is that they are sick people with TB, and one has to be very careful. Signs of fear were also expressed.

A community member of forty-seven years old and four years of schooling expressed his view:


*“I see that person as a human being, as a colleague of us… But the only thing is that I know he or she is a TB patient, he is infected, and I know that he is someone who is sick. And I need to be careful not to be infected by his sickness.”*
—Non-TB participant #3

Another community participant, thirty years old and with thirteen years of schooling, said:


*“As for me, I will not dislike the person, I will be very careful not to get close to that person because TB is very dangerous you can die.”*
—Non-TB participant #4

A community member of forty-five years old and eight years of schooling continued:


*“I mean, I can die; it is serious korseh kheka (long-lasting cough). I will not hate the person but please l will never get near him, it is a bad disease.”*
—Non-TB participant #5

A community member of twenty-seven years old and twelve years of schooling said:


*“What disturbs me most about the TB patients is that we know TB can kill, and you know it is also contagious and we know that as he is in the community, he might infect us. His own life is also at risk, so these are some of the things that disturb me.”*
—Non-TB participant #6

#### 3.2.3. Theme #3: Social Consequences of Stigma towards TB

When asked why people stigmatize them, it emerged that the major cause of stigmatization was the fear of being infected and the dangers associated with the infection. This conclusion was drawn after views were analyzed from non-TB FGD participants regarding whether they will share the same cup with a TB patient, stay in the same house with a TB patient, or sit by a TB patient in a car.

On sharing the same cup, a respondent of fifty-one years old with no education said:


*“… Because this person (referring to TB patient) is already sick and being in the same house, I would not like to drink from the same cup. We know that as a TB patient you are admitted to the ward for some time and before discharge, I don’t like to be admitted due to getting infection from him/her, and I may die.”*
—Female non-TB participant #1

Concerning staying in the same house, this was said by forty-seven years old and thirteen years of schooling participant


*“In the same house, there is nothing wrong with that; it is only when you sleep together and the person coughs that you will get it, but if you are in different rooms staying in the same house, there is no problem with that.”*
—Male non-TB participant #2

Sitting by a TB patient in a car


*“(Respondents shake their heads in sign of disagreement) As we already said because it is contagious, and we have been educated that when the person sneezes, coughs or yawns he is bringing it out, and you are likely to be infected, so I will not be comfortable to sit in a car with a TB patient.”*
—Female non-TB participant #2

However, the FGD also indicated that non-TB participants were willing to stay in the same room with their husband, who was infected with TB, so long as he was undergoing treatment.


*“(Respondents laughed) If my husband has TB, my relationship will be normal because he is my husband, and if I do not stay because of TB, he will divorce me, so we will stay together and we will manage it.”*
—Female non-TB participant #3

TB patients felt awful when they were stigmatized. They realized that they should not avoid other TB patients after their recovery. One of the females explained that:


*“I felt bad when my people shunned my company, so I will not do that to others, and I will advise the person to come to the hospital for treatment.”*
—Female TB patient #1

Despite the fact that some non-TB participants, to some extent, stigmatize TB-infected people, they did not feel comfortable doing that, but because of a lack of education and knowledge of the stages of infection of TB, they had no choice.


*“I don’t feel comfortable. It is not good to stigmatize something like that because you may also be infected one day and if somebody stigmatizes you or avoids you, you will not like it that way; so when you do that to them, it is not the best way.”*
—Non-TB participant #1

It was, however, admitted in the discussions among the non-TB participants that it was bad to stigmatize TB patients since this could worsen their plight; it could even kill them, but rather TB patients should be supported, and compassion showed to them. 

A participant of fifty-one years old and four years of schooling said:


*“I should help them for their sickness at least maybe if somebody is staying alone without relatives then you know that this person is in my community, and he is having this condition you can help in his washing of things, bowls and maybe provide him some meals. So, if somebody is having something like that I should be able to help that person.”*
—Non-TB participant #2

Another person continued and said:


*“Stigmatization too won’t help. Stigmatization will rather worsen the case. If we stigmatize them, they may not feel well they will think as if they are not human beings, and we are rather harming them the more and at times when you stigmatize them like that they are not of themselves, and they can die out of the stigma.”*
—Non-TB participant #3

Social Effect of TB

There were various effects that TB was said to have on individuals in the community, which really brought about some changes in the households. 

Effect on Marriage

There were different perceptions about the marriage of TB-infected patients to their daughter/son. However, men were more confident than women in their children’s marriage in society despite their TB status.

A patient of thirty-eight years old with no education said in FGD:


*“The one who would be marrying my daughter/son will be scared because I have the TB infection.”*
—Male TB patient #2

A female participant said in another FGD:


*“There will not be any problem because I gave birth to him/her before the sickness.”*
—Female TB patient #2

Families of non-TB participants will also be comfortable marrying a son/daughter of a TB patient who has been on treatment for six months because they believe he/she will not infect them with the sickness. 

A female participant of forty-five years old and eight years of schooling said during FGD:


*“As there is no problem because now we know TB is curable, so he has been cured and there is no problem.”*
—Female non-TB participant #3

Effects on Household Income

The comments from the discussions revealed that participants believed a TB-infected patient’s status would have no effect on household income or income-generating activities. An employed TB patient said during our discussion that, 


*“I am a security man and as at now I have put somebody at my place to work so it is not affecting anything.”*
—Male TB patient #3

Again, continued another TB patient:


*“No, I don’t think there is going to be a problem in my family, because they will still work at the level that they can, they should also work and change for the better.”*
—Male TB patient #2

The females, however, harbored some worries. This could be because they were neglected mostly by their family members. They were viewed as liabilities because they could not help themselves.

A patient was prompted to say during the discussion,


*“They said, I should not work, and I am now wondering how I am going to feed the children.”*
—Female TB patient #4

Another TB patient continued:


*“As for me, it is hard; I can’t work on the farm and get money and when l go to home, my family cannot help me, and my husband is poor.”*
—Female TB patient#5

This respondent’s demeanor was very sad, and she sounded helpless about her situation. She is rather better off in the hospital than being discharged after she recovers from her sickness. This situation demoralized her psychologically and might affect her recovery.

#### 3.2.4. Theme #4: Perceptions about TB Diagnostic Tests and Taking Treatment

Generally, TB patients expressed no fears in taking their treatment since they believed that when they take their medications well, they will be cured of the TB infection. It was also stated that there were no reactions associated with taking the drugs. 


*“No, when I take the drugs, I have not had any reaction the drugs are good. At first when I eat, I vomit but when I started taking the drugs, I am now okay.”*
—Female TB patient #5

Another patient continued:


*“Before I had vomited a lot, but when the nurse gave me these drugs in the red box l didn’t vomit as I use to, I take it every day.”*
—Female TB patient #6

Another patient also expressed himself by saying: *“I don’t have any fear; I feel better now as I have started taken the drugs.”*—Male TB patient #3 

TB patient continued in the discussion: *“The way I was coughing is not the same, previously I was also very slim, but now I am better.”*—Male TB patient #5

These male participants demonstrated a positive attitude which shows that after following the treatment course, they will surely recover and be free from TB and not be stigmatized.

Voluntary Counseling and Testing of TB

In discussion with the non-TB participants, participants’ views were solicited to discover whether voluntary counseling and testing of TB would be accepted in the community. The general view was that communities would consider voluntary counseling and testing of TB because this would help them learn how to prevent themselves from being infected and, if they are infected, where to get treatment.


*“… Voluntary counseling and testing on TB, yes I will consider it because maybe I am not infected yet, and if they counsel me, they may tell me what to do if I am not infected. They may tell me how to prevent myself or if I am infected, they will tell me where to go, and they will treat me to come out of the TB, so I will consider voluntary counseling and testing because it will help me a lot.”*
—Non-TB participant #3


*“… It also helps you to know your immunity, how strong it is. Maybe you can be harboring it for so many months, and you wouldn’t know you may think it is common cold. So, when you cough there is nothing wrong with you and when you encourage that you will go in for it and it will help you know how your system is moving, and you can also prevent yourself from getting it.”*
—Non-TB participant #2

## 4. Discussion

### 4.1. Perception of Signs and Symptoms Observed by TB Infected Person 

TB infection was described by the patients as pain in the ribs, chest pain and difficulty in breathing, and sometimes vomiting after meals. These findings justify the previous study findings in the Ntcheu district of Malawi, where they reported that TB patients felt pain inside their chest during breathing and sleeping on one side [30]. Most TB patients had their diagnosis at the hospital after a radiography test. Some participants suffered from persistent illnesses before being diagnosed as TB-infected patients. This delay allows the spreading of this infectious disease to the family and the community [8,10]. Some TB patients were delayed attending the hospital even though treatment for TB infection was available free of charge in all health facilities. The reasons given for recourse to the hospital for treatment included the fact that there was no other place of treatment known to them. Similar findings were observed elsewhere; patients were not aware of the free treatment facilities provided by public healthcare facilities [31]. Most of them were, however, advised by family members and relatives to go to the hospital. This could mean that some people still do not have adequate knowledge of the symptoms of TB infection and where to receive treatment for the infection. Patients should be aware of the signs and symptoms of TB and attend the nearby hospital or clinic without any delay; however, it was found not to be the practice in many previous studies [30]. One study found that participants who had good knowledge suffered significantly less with TB stigma [32]. This could go a long way to reducing the infection of more people in the communities since patients would have knowledge of the early signs of symptoms, thus responding by receiving early treatment, thereby reducing the early spread of infection to other community members. It is recommended that persons with a cough for two to three weeks should seek medical care since they could be infected with TB [33]. However, individuals from high-prevalence areas, households, and close contacts should undergo TB screening regardless of cough duration [34]. 

There was also a lack of knowledge about the source of infection irrespective of participants’ years of schooling, as almost all the participants who were infected could not tell exactly where they were exposed to the infection. Even though one participant suspected that she was exposed through eating with her family members, it was not justifiable since records of infection by any of the family members could not be established. No one was, however, to be blamed for TB as the cause of infection was like a miracle since all participants believed it was a sickness that came on its own. Healthcare-provider–patient communication can improve knowledge and reduce the TB stigma [35]. 

### 4.2. Attitudes towards TB Patients before and after Diagnosis as Being Infected 

In our study, a male TB patient mentioned that he would not be a burden to the family and that his inability to work would not affect the household income. Based on what the TB patient explained, he would not be a burden on his family since, culturally, the family is expected to contribute financial support to him whilst he is unemployed. This could be attributed to the nature of the income-generating activities as most of the participants were farmers, and it did not need any great skill in order to engage in it; hence family members could take that up easily. However, in reality, TB made the patients poor due to loss of work and work capacity. Additionally, treatment expenditure worsens their scarcity and makes the people more vulnerable to avert diagnosis and receive adequate treatment [36]. TB patients with less social support and having moderate or severe disease intensity experienced more stigma and suffered more psychological distress that hampered their quality of life [35,37]. Non-TB participants were comfortable exposing themselves to the risk of becoming infected in order to maintain their relationship with their close relatives, such as their husbands, but not any other ordinary TB patient, where they admitted emphatically that they could not risk being exposed to the infection.

The worry of the female participants, who, from the findings, were the most neglected. They were left alone to take care of their children, and due to the difficulty associated with that, they were also advised not to do tedious work again. Moreover, they perceived heredity as more stigmatized compared to men [38]. Hence compassion should be shown to female patients infected with TB.

### 4.3. Social Consequences of Stigma towards TB

According to the findings, men and women whose parents or relatives are known to be infected with TB would not have any difficulties getting married. Though one female TB patient stated that the one going to marry her daughter/son would be scared of getting infected, the general perception was that it is the parent who is infected; hence there was no reason to refuse marriage. This was confirmed in the discussions with the female non-TB participants. Culturally, TB-infected patients were avoided by their family members and the community due to the stigmatization of being infected by them. Inappropriate health education messages by the media aggravate the tuberculosis stigma that isolates TB patients, fails them in financial crises, delays diagnosis, reduces treatment adherence, and leads to poor outcomes/prognoses [39,40]. Female TB patients felt the stigmatization more than their male counterparts. The findings also indicated that it was mostly females who were neglected by family members rather than males. The males generally had their family members, community members, and friends visiting them, which was not the case with female patients. This could be attributed to cultural beliefs that still exist in some parts of developing countries, such as in Ghana in Bolga, the Upper East Region, where the man is the head of the house. Traditionally men see themselves as the head of the household, and this concept has not changed. Hence, society did not see an infected woman as useful to their family anymore. She was seen as an outcast, unlike men. The same findings were found in previous studies where females were more stigmatized compared to males [35,40].

Discrimination was obvious where men were able to solicit help from family members whilst women were denied these privileges. Women were more vulnerable to receiving support from their spouses compared to their counterparts [38]. However, it is imperative to provide physical and emotional support to TB patients without discrimination in receiving treatment [29]. The major reason for stigmatization was the fear of being infected. When TB patients were stigmatized, community members expressed a feeling of unhappiness. The reason given was that stigmatization would only worsen the plight of the patients and, in serious cases, could even kill the patient. One study conducted in community people of Nigeria observed that though most of the participants were conscious of TB, nonetheless their level of knowledge, attitude, and practice were poor. Participants of 22.7% had TB stigma, and less than half (41%) had a negative attitude toward TB patients. Many (63.6%) of them had the perception not to show up compassion and desire to help TB patients. Moreover, 64.3% opined TB patients should not be employed [32].

### 4.4. Perception of Diagnostic Tests and Taking the Drugs 

The findings suggested that the TB patients did not have fears about taking the TB medication. It is believed that when they take their drugs, they will be cured. There were no reactions mentioned to be associated with taking the drugs, and the drugs were said to be efficacious. There was a general acceptability of voluntary counseling and testing for TB by communities; this was indicated in discussions with non-TB participants. Voluntary counseling and testing for TB will help community members know how to prevent being infected and, if they are infected, where treatment can be obtained. The finding showed that most of the TB-infected patients reported and started treatment after a month or more which would have been enough time to infect several others. In order to achieve a 90% reduction of TB incidence by 2035, it is imperative to educate community people about the symptoms of TB for early case detection and access to health care facilities [41]. TB stigma caused significantly more delay in care seeking among the persons with higher stigma compared to those who had less [42]. TB stigma also hampered adherence to DOTS therapy [43]. One study observed that compared to educated patients, uneducated patients made for a long delay in detecting TB, and it should be considered in the NTP [44]. 

### 4.5. Study Limitations

In our study, we used FDGs to explore the participants’ perceptions, attitudes, experiences, and opinions on TB stigma has its documented limitations. Study participants were included from only one municipality; participants from other areas might have different perceptions, attitudes, experiences, and concerns that might influence the study findings. Therefore, the generalizability of the study findings could not be achieved due to the nature of the study. We also used only the FGDs method to find out an in-depth understanding of social issues; hence, methodological triangulation was absent, and we could not overcome the intrinsic bias. However, while forming the groups for FGDs, we purposively selected TB-infected patients and non-TB patients, which helped us triangulate the findings in terms of the source of participants. 

## 5. Conclusions

Stigmatization is the worry harbored by most TB patients and community participants. Females are more pronounced to TB stigma due to the traditional cultural beliefs that exist in the community. Moreover, females are less empowered in decision-making for TB management due to their economic vulnerability. TB patients were unhappy being stigmatized, and community members frowned upon it. They considered that TB patients were also human beings, and stigmatization would only worsen their plight. It is suggested that the World Health Organization should encourage developing countries to focus on educating their population more about stigmatization, especially regarding the dignified treatment of people with this condition. Regional Directorate of the Ghana Health Service in Bolga should heighten Advocacy, Communication, and Social Mobilization in the Bolgatanga Municipality to allay stigma as a means of promoting health and decreasing TB transmission among the general population, especially women. TB education should be factored into every facet of endeavor to dispel the stigma against TB-infected individuals.

### 5.1. Future Research Direction 

As Ghana is still considered a TB-endemic country, an ethnographic study could be needed to explore more insights into underlying factors associated with the stigma of TB-infected people.

### 5.2. Implementation of Research Findings

In our study, we observed that TB patients and community people were stigmatized. Females were more vulnerable to harmful consequences such as discrimination, social exclusion, and economic difficulties. Continuous exploring and understanding of TB stigma would help in designing a more comprehensive approach to the management of TB in the NTP. Effective public–private partnerships should be employed with predominant stakeholders and upsurge awareness for TB-associated inequalities in the communities. 

## Figures and Tables

**Table 1 ijerph-19-14998-t001:** **Discussion** **guideline for FGDs.**

Introduces a picture of a TB patient to the group (7 males and 6 females TB patients).
Moderator: Opinions, knowledge and experience of stigma as a TB Patient?
1. Disclosure: Whom did you tell about your TB status? Probe after response, 2. How did you find out about your TB status? 3. What did your friends do when you told them that you have TB? Please explain 4. When did you first seek treatment? Please explain 5. How do you feel as a TB patient in your community? 6. Why do you think your wife or husband contracted TB? 7. What were the changes after your wife/husband found that you have TB? 8. What are your worries as a TB patient? Probe after response 9. Who do you blame for your TB infection? 10. Would your wife or husband’s TB condition affect the family daily income? 11. What is your fear of taking the prescribe drugs? 12. Would your son or daughter have any difficulty in getting married in the community? Please explain 13. How do people in the community feel towards you? Is there any more information anyone wants to share with the group?
General Population FGD (6 persons not infected)
Moderator: Opinions, Knowledge and experience of persons not infected
1. How do you feel about TB patients? Probe after response 2. Would you drink from the same cup with a TB Patient? 3. Would you still keep a friend who had just being diagnosed of TB? 4. What are your worries when you see a TB patient? Probe after response 5. Would you live in the same house with a TB patient? 6. Would you allow your brother or sister to marry a person who has been treated for six months for TB? Please explain 7. How would you feel towards your husband or wife who has TB? Probe after response 8. Would you sit beside a TB patient in a car? Please explain? 9. Would you consider Volunteer Counseling and Testing? Probe after response Is there any more information anyone wants to share with the group?

**Table 2 ijerph-19-14998-t002:** **Meaning units,** **codes, sub-themes, and themes.**

Meaning Units	Codes	Sub Themes	Themes
Before I came to the hospital, I vomited when I ate. I also complained of rib and chest pains. So, it was my son who even told me that I was sick and when we came to the hospital, they took radiography and then told me it was TB (female TB patient).	Visit to the hospital, vomiting, rib and chest pain, radiography showed TB.	Vomiting, chest pain, rib pain, chest radiography.	Signs and symptoms observed by TB infected person.
When it started it didn’t keep long, and I took medicine, and it stopped. It came again and I took medicine again, and it stopped for the third time. I didn’t know what was wrong with me, so I came to the hospital, and they told me it was TB (male TB patient).	Medication stopped for three times, hospital informed TB.	Medication stopped, hospital informed.	Signs and symptoms observed by TB infected person.
I see that person as a human being, as a colleague of us… But the only thing is that I know he or she is a TB patient, he is infected, and I know that he is someone who is sick. And I need to be careful not to be infected by his sickness (non-TB participant).	Fear of infection, person sick with TB, be careful, avoid infection, not get closer.	Fear of infection Attitudes of friends Relatives and community.	Attitudes towards TB patients before and after diagnosis, as being infected.
I will not get closer to that person because I know TB is something that is contagious and dangerous so if I get close to the person, he/she will infect us with the sickness he/she has gotten (non-TB participant).	Fear to be infected, TB contagious, dangerous, not get close.	Fear of infection, TB contagious, dangerous, not get close.	Attitudes towards TB patients, fear of being infected.
(Respondents shake their heads in sign of disagreement) As we already said because it is contagious, and we have been educated that when the person sneezes, coughs or yawns he is bringing it out, and you are likely to be infected, so I will not be comfortable to sit in a car with a TB patient (female non-TB participant).	Knowledge about TB, contagious, infected by sneezing, coughing, yawning, not sit in same car.	Contagious, fear of infection Knowledge about TB.	Social consequences of stigma towards TB.
Of course yes, it is the matter of staying in different rooms but not divorcing him (female non-TB participant).	Knowledge about TB, contagious, infected by sneezing, coughing, yawning, not sit in same car.	Contagious, fear of infection Knowledge about TB.	Social consequences of stigma towards TB
No, when I take the drugs, I have not had any reaction the drugs are good. At first when I eat, I vomit but when I started taking the drugs, I am now okay (female TB patient).	Drugs no reaction, feels good with drugs, no vomiting.	No fears No sign of any reaction.	Perception about diagnostic test and taking the drugs.
Before I vomited a lot, but when the nurse gave me these drugs in the red box l didn’t vomit as I use to, I take it every day (female TB patient).	Drugs no reaction, feels good, no vomiting, red drugs take everyday.	No fears, no sign of any reaction.	Perception about taking the drugs.

**Table 3 ijerph-19-14998-t003:** **Socio-Demographic** **characteristics of the participants.**

Three Focus Group Discussion and Total participants N = 19		
	Male	Female
1st FGD	0	6
2nd FGD	7	0
3rd FGD	3	3
Total	10	9
Age (mean) (years)		
1st FDG	35	
2nd FDG	41	
3rd FDG	43	
Marital status	N	
Married	9	
Single	4	
Divorced	3	
Widow	3	
Education	N	
No education	4	
Primary education (primary 5 level)	3	
Junior High School (8 years of schooling) Senior High School and above (9 years and above)	8 4	
Profession	N	
Farmer	11	
Trader	8	
Religion	N	
Traditionalists	9	
Christian	5	
Muslims	5	

N, number.

## Data Availability

The data presented in this study are available on request from the corresponding author. Full data are not publicly available due to privacy restrictions.

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
