# Peer review of "Perceptions, Attitudes, Experiences and Opinions of Tuberculosis Associated Stigma: A Qualitative Study of the Perspectives among the Bolgatanga Municipality People of Ghana"

_ijerph, 2022, doi:10.3390/ijerph192214998_

Round 1

Reviewer 1 Report (Previous Reviewer 4)

The authors adequately addressed my comments.

Author Response

Date:   September 21, 2022
Subject: Re-submission of the Manuscript ID: ijerph-1834479

Thank you very much again for your further consideration of my manuscript for possible acceptance.

Best regards,

KATM Ehsanul Huq, MBBS, DTM, MSc, PhD
Health Sciences Major
Graduate School of Biomedical and Health Sciences
Hiroshima University, Japan
Mobile:080 6266 8578

Reviewer 2 Report (New Reviewer)

Overall I consider the paper in good shape, but some more care is needed when discussing, evaluating and contextualizing the findings, and sometimes the language needs a little more attention.

Introduction

The introduction starts with current trends, whereas the study is done in 2012-3. I feel the introduction should start with trends around 2010. Now when reading from the introduction to the materials and methods you feel disappointment, and ask yourself if the study is sufficiently up to date. I believe the findings are sufficiently stable, but the authors need to make this argument stronger, preparing it in the introduction.

Materials/methods: time and context of the study

When explaining when the study was done (Line 110), they authors need to argue not much has changed over the last 10 years regarding TB stigma fears, or knowledge, to make the study still relevant today.

Also relevant might be to compare the level of education and medical knowledge about TB in the region as compared to Ghana as a whole, or Africa in general.

Findings: relevance of schooling

During the discussion of the findings years of schooling are mentioned, and this might be relevant, but it has nowhere be explained why, or is it linked to interpretations. Are better educated patients different in their approaches or attitudes, or not. As a reader you do not get an answer on this question.

Discussion: normative statements and contextualization

I recommend to put all normative statements/implications in one last paragraph. They do not follow from the findings. These norms also need a better argument (for instance line 702; 727-8)

A normative evaluation might be better integrated in section 7.

Contextualizations:

The Nigeria study is not well compared to the Ghana context. Line 733-37 need more attention

Caste: explain you discuss a finding from India, and relate this finding to Ghana (Line 749).

Study limitations:

Time aspect needs attention here (findings from 2012-2013). How are things different now?

Future research – connect ideas for future research to the study limitations, present ideas to overcome them

Language:

Check line 596; 617; 684-6; 731; 756; 765 (and more…).

Author Response

Date:   September 21, 2022
Subject: Re-submission of the Manuscript ID: ijerph-1834479

Thank you very much again for your valuable comments and further consideration of my manuscript for possible acceptance.

We addressed all the reviewer’s comments point-by-point and all the changes are marked by “Track Changes” within the revised manuscript.

Best regards,

KATM Ehsanul Huq, MBBS, DTM, MSc, PhD
Health Sciences Major
Graduate School of Biomedical and Health Sciences
Hiroshima University, Japan
Mobile:080 6266 8578

Review Report Form 2

Comments and Suggestions for Authors

Introduction

The introduction starts with current trends, whereas the study is done in 2012-3. I feel the introduction should start with trends around 2010. Now when reading from the introduction to the materials and methods you feel disappointment, and ask yourself if the study is sufficiently up to date. I believe the findings are sufficiently stable, but the authors need to make this argument stronger, preparing it in the introduction.

Responses: Many thanks for your valuable comments. According to your advice, we have incorporated references in page: 2, line: 70-76.  

Materials/methods: time and context of the study

When explaining when the study was done (Line 110), they authors need to argue not much has changed over the last 10 years regarding TB stigma fears, or knowledge, to make the study still relevant today.

Responses: Many thanks for your excellent suggestion. We have incorporated this statement in page: 3, line: 113-115.

Also relevant might be to compare the level of education and medical knowledge about TB in the region as compared to Ghana as a whole, or Africa in general.

 Responses: Many thanks for your valuable comments. We added the education condition of Ghana in page: 3, line: 120-122.

Findings: relevance of schooling

During the discussion of the findings years of schooling are mentioned, and this might be relevant, but it has nowhere be explained why, or is it linked to interpretations. Are better educated patients different in their approaches or attitudes, or not. As a reader you do not get an answer on this question.

Responses: Thank you very much for your important observation. We have incorporated the overall interpretations in page: 19, line: 694-695.

Discussion: normative statements and contextualization

I recommend to put all normative statements/implications in one last paragraph. They do not follow from the findings. These norms also need a better argument (for instance line 702; 727-8)

Responses: Thank you very much for your constructive comment. We have incorporated references for comparing with other study findings in page: 20, line: 724-725 and page: 20-21, line: 750-751.

A normative evaluation might be better integrated in section 7.

Responses: Thank you very much. According to your advice, we incorporated it in section 7, page: 22, line: 818-820.

Contextualizations:

The Nigeria study is not well compared to the Ghana context. Line 733-37 need more attention

Responses: Many thanks for your comments. We revised the statement in page: 21, line: 755-761.

Caste: explain you discuss a finding from India, and relate this finding to Ghana (Line 749).

Responses: According to your suggestion, we revised in page: 21, line: 775-777.

Study limitations:

Time aspect needs attention here (findings from 2012-2013). How are things different now?

Responses: Thank you for your comments. We already added this time aspects in the ‘Introduction’ part in page: 2, line: 71-77.

Future research – connect ideas for future research to the study limitations, present ideas to overcome them

Responses: We revised the ‘Future Research Direction’ accordingly in page: 22, line: 811-812.

Language:

Check line 596; 617; 684-6; 731; 756; 765 (and more…).

Responses: We corrected the language of the above-mentioned lines.

Reviewer 3 Report (New Reviewer)

Firstly, I thank Editorial Committee for the opportunity to review this manuscript.

The authors deal with an important issue. Understanding the perceptions, attitudes, and experiences regarding stigma among people with tuberculosis is relevant.

This study focuses on the exploration of the perceptions of patients in Bolgatanga Municipality of Ghana.

The proposed manuscript adequately meets the purposes of the journal. In addition, the paper presents some interesting findings.

Below I suggest some recommendations suggestions improve the quality of the manuscript:

The aim is clearly stated - Abstract: “The objective of this study was to explore the perceptions, attitudes, experiences and opinions about stigma related to TB among adults infected with TB and adults who were not infected with TB”; however, it is not in line with the aims presented in heading - 1.2. Purpose – “We conducted this study was to explore the perceptions, attitudes, experiences  and concerns of TB patients as well as community members to gain a deeper understanding of the phenomenon of TB stigma, how it is manifested, and specific behaviors and consequences associated with TB” and again in heading 1.3. significance “This is the first study to explore the TB stigma, worries and experiences of TB patients, attitudes of community members towards TB patients among the people of Ghana.” In the heading Study limitations, authors refer that “In our study, we used FDGs to explore the participants’ perceptions, attitudes, experiences and concerns on TB stigma has its documented limitations. Throughout the manuscript, the aims should not differ.

The title should also be in line with the study aim – “Perceptions, Attitudes and Consequences of Tuberculosis As- 2 sociated Stigma: A Qualitative Study of the Perspectives 3 among the Bolgatanga Municipality People of Ghana”.

The abstract should be a single paragraph and should follow the style of structured abstracts but without headings. Please remove the headings Background, Methods, Results and Conclusions from the abstract.

Line 71 – Authors present HIV for the first time. It should be defined.

Line 208 – Authors stated, “This test enabled the researcher to understand the interview techniques and data collection methods before the study started. We also found out everyone’s knowledge about TB”. Pretesting will help us determine if participants understand the questions as well as if they can perform the tasks or have the information that questions require. That is the information that the authors should provide in the manuscript. Please state if the guide was sufficiently explicit, objective, comprehensive, and present or not questions that could be ambiguous or equivocal.

Heading 2.1.5 Trustworthiness - It could be structured in line with trustworthiness referred by Nowell, Norris, White, and Moules (2017) through credibility, transferability, dependability, and confirmability. (Nowell, Lorelli S., Jill M. Norris, Deborah E. White, and Nancy J. Moules. "Thematic Analysis." International Journal of Qualitative Methods 16, no. 1 (2017): 160940691773384.

Following COREQ (Consolidated criteria for Reporting Qualitative research) Checklist would be helpful to develop the “The study” section (methods) as several items that should be included in reports of qualitative research are missing. Overall, please go back to COREQ and check what is missing from your article. You need to address each item in that list. For example, 

Ø What were the moderator's credentials?

Ø What experience or training did the researcher have?

Ø How did the authors evaluate data saturation?

Ø How many people refused to participate in the study? 

4.5. Study limitations – Authors should address the limitations derived from the methodological choices (study design, sampling, data collection instrument …)

Author Response

Date:   September 21, 2022
Subject: Re-submission of the Manuscript ID: ijerph-1834479

Thank you very much again for your valuable comments and further consideration of my manuscript for possible acceptance.

We addressed all the reviewer’s comments point-by-point and all the changes are marked by “Track Changes” within the revised manuscript.

Best regards,

KATM Ehsanul Huq, MBBS, DTM, MSc, PhD
Health Sciences Major
Graduate School of Biomedical and Health Sciences
Hiroshima University, Japan
Mobile:080 6266 8578

Review Report Form 3

Comments and Suggestions for Authors

The aim is clearly stated - Abstract: “The objective of this study was to explore the perceptions, attitudes, experiences and opinions about stigma related to TB among adults infected with TB and adults who were not infected with TB”; however, it is not in line with the aims presented in heading - 1.2. Purpose – “We conducted this study was to explore the perceptions, attitudes, experiences  and concerns of TB patients as well as community members to gain a deeper understanding of the phenomenon of TB stigma, how it is manifested, and specific behaviors and consequences associated with TB” and again in heading 1.3. significance “This is the first study to explore the TB stigma, worries and experiences of TB patients, attitudes of community members towards TB patients among the people of Ghana.” In the heading Study limitations, authors refer that “In our study, we used FDGs to explore the participants’ perceptions, attitudes, experiences and concerns on TB stigma has its documented limitations. Throughout the manuscript, the aims should not differ.

Responses: Many thanks for your valuable comments. We revised the aim in purpose, significance and in limitations (page: 3, line: 98-102 & 105-107; page: 21, line:778). 

The title should also be in line with the study aim – “Perceptions, Attitudes and Consequences of Tuberculosis Associated Stigma: A Qualitative Study of the Perspectives among the Bolgatanga Municipality People of Ghana”.

Responses: Thank you very much. According to your advice, we changed the title in page 1.

The abstract should be a single paragraph and should follow the style of structured abstracts but without headings. Please remove the headings Background, Methods, Results and Conclusions from the abstract.

Responses: Many thanks. According to your advice, we deleted the headings from the abstract.

Line 71 – Authors present HIV for the first time. It should be defined.

Responses: Thank you very much. We have elaborated it in page: 2, line: 73.   

Line 208 – Authors stated, “This test enabled the researcher to understand the interview techniques and data collection methods before the study started. We also found out everyone’s knowledge about TB”. Pretesting will help us determine if participants understand the questions as well as if they can perform the tasks or have the information that questions require. That is the information that the authors should provide in the manuscript. Please state if the guide was sufficiently explicit, objective, comprehensive, and present or not questions that could be ambiguous or equivocal.

Responses: Thank you for noting this. We have revised it accordingly in page: 6, lines: 218-227).

Heading 2.1.5 Trustworthiness - It could be structured in line with trustworthiness referred by Nowell, Norris, White, and Moules (2017) through credibility, transferability, dependability, and confirmability. (Nowell, Lorelli S., Jill M. Norris, Deborah E. White, and Nancy J. Moules. "Thematic Analysis." International Journal of Qualitative Methods 16, no. 1 (2017): 160940691773384.

Responses: Thank you for your suggestion. We have added the statement and reference in page: 6, line: 241-242.

Following COREQ (Consolidated criteria for Reporting Qualitative research) Checklist would be helpful to develop the “The study” section (methods) as several items that should be included in reports of qualitative research are missing. Overall, please go back to COREQ and check what is missing from your article. You need to address each item in that list. For example, 

Ø What were the moderator's credentials?

Ø What experience or training did the researcher have?

Ø How did the authors evaluate data saturation?

Ø How many people refused to participate in the study? 

Responses: Thank you for your valuable suggestion. We have incorporated in page: 6, line: 243, page: 4, line: 161, page:7, line: 288-289, and page: 4, line: 133-135. We have reported our study following the COREQ checklist and the COREQ checklist has been provided as a supplementary file (Supplementary file 1: COREQ checklist).

4.5. Study limitations – Authors should address the limitations derived from the methodological choices (study design, sampling, data collection instrument …)

Responses: Thank you for your suggestion. We have elaborated study ‘limitation section’ addressing the issues on study design, sampling and data collection instrument in the revised version of the manuscript (page: 21).

Round 2

Reviewer 3 Report (New Reviewer)

I believe the authors have responded adequately to my comments and made the necessary changes. Therefore, I consider that this manuscript is fit to be published.

This manuscript is a resubmission of an earlier submission. The following is a list of the peer review reports and author responses from that submission.

Round 1

Reviewer 1 Report

The manuscript describes a qualitative study exploring the "lived experiences, perceptions and opinions about stigma related to TB" among adults in Ghana. I congratulate the authors for submitting an interesting study but cannot recommend the manuscript for publication at this time due to the following: 

1) There are consistent grammatical errors throughout the manuscript - across all sections. The manuscript would benefit from several more rounds of editing and careful revision; 

2) I have concerns over the authors' use of the proposed thematic content analysis. While I agree that this method would be appropriate, I was not convinced by the methods and analysis section the authors applied this technique correctly. For example, even in the abstract, it was not clear what the themes of the analyses were. How were codes developed? How were themes gathered from the careful review and analysis of the focus group interviews? Was inter rater reliability established? If no, why not? 

3) I also had questions regarding the recruitment of participants and how the focus groups were conducted. 

4) The authors introduced additional collaborators in the manuscript, such as an "external supervisor," whom assisted in reviewing the methods and analysis. Does this supervisor have the qualifications to assess qualitative methods? If so, that needs to be mentioned. 

5) Finally, the manuscript would greatly benefit from clearly aims in the introductory section. In addition to exploring perceptions, attitudes, and concerns, why did they authors chose not to compare such constructs between the two groups of participants (TB vs. non-TB)?

Abstract: 

  1. Should describe the themes from your analysis here
  2. The second second in line 23 should be rewritten as it is unclear what the authors are trying to say. Are they trying to stay stigma is a barrier to seeking treatment and adequate care? 
  3. The objectives should be made clearer. Are the authors exploring AND comparing or only exploring the perceptions, attitudes, etc. Also, the objectives here mention "lived experiences" and "opinions" which are not mentioned in the introductory section. Therefore, do not seem to completely overlap. 
  4. There are several grammatical issues throughout this section alone. For example, Line 38 should say "worsen" and not "worsening."

Introduction:

  1. This section was informative, however, I would recommend the authors describe why their study is novel or why their study is filling an existing a knowledge gap. Furthermore, what have previous qualitative studies in this area identified? Or, is your study one of the first few?
  2. While stigma is mentioned quite a bit throughout the paper, the introduction should strengthen the description of how stigma manifests itself and acts as a barrier to seeking care. 

Materials and Methods:

  1. Why did the authors choose to conduct focus groups or other qualitative methods? Were men and women a part of the same focus group or were they separated based on gender and sex? 
  2. Define the community you sampled from. 
  3. It is great the authors harnessed the influence of community leaders. Without identifying them, what were their roles in the community? Why were those leaders selected in particular? Were they involved in the focus groups given the focus groups were conducted in their homes? 
  4. It is unclear how having the focus groups in the homes of community leaders gave the authors "access to their opinions and attitudes about how they felt about TB patients." 
  5. How long were the focus groups?
  6. Is the interview guide a supplemental material that reviewers can read? What types of questions were there? 
  7. Was a pre- or post-interview questionnaire given to the participants? If not, how did the authors obtain information such as participants knowledge on specific topics? For example, the authors wrote "we also found out everyone's knowledge about TB." This part of the section should be made clearer. Was a specific question asked about this orally? 
  8. Again, the relationship between the authors and the external supervisor should be made clear. Is this external supervisor a qualitative researcher or has some familiarity with the study? Was he/she/they approved by IRB or another entity to listen to audio, identifiable interviews that the authors collected? 

Data Analyses:

  1. What are some example codes and sub-themes?
  2. Given that you are using qualitative content analysis, was there any use of an existing framework, model, or theory to help guide the development of codes and themes?
  3. Was a codebook established and modified until the authors reached agreement?
  4. The process of how the authors obtained their themes need to be elaborated. 

Results:

  1. Was the data described in the first paragraph gathered from a questionnaire or short survey?
  2. Is it possible to also match the quotes to the participants' ages as individuals across age groups might have different lived experiences?
  3. The use of sub-themes was hard to distinguish in the manuscript. Consider bolding them or italicizing. I am not familiar with the authors' data, but am wondering if theme 3 is more about the social consequences of stigma towards TB rather in addition to "reasons for stigmatization." This is my assessment based on the quotes and subthemes provides. 
  4. I would recommend the authors revisit the development of the themes. Overall, the aims also need to be revisited and the themes should ideally be derived to align with those aims (e.g., attitudes, perceptions, etc.). The aims mentioned in the introduction and abstract may also tie in the role of stigmatization as it comes up often in the manuscript.  
  5. Please also proofread this section. 

Discussion: 

  1. What additional limitations are there? 
  2. Was triangulation used to strengthen your study?
  3. How did you know if you reached data saturation? In other words, was 19 participants enough? How did the authors determine that?

Reviewer 2 Report

The paper is interesting and it addresses a quite interesting topic. However, some issues should be addressed.

  1. The title does not include attitudes even though it is a main theme in your study.
  2. Stigma, perception, attitude and opinion are different concepts . These concepts should be defined and sufficient literature review about stigma, attitudes, perceptions, opinions related to TB stigma should be provided in the introduction.
  3. Research participants’ attitudes towards TB patients are presented in your paper as precautionary/protective measures and may not be described as stigma. Generally, stigma is associated with shame, isolation and fear. These three characteristics of stigma should be illustrated and elaborated.
  4. The present simple should be used consistently in your paper as there is a shift of using past simple.
  5. There is a reference to a supervisor of this research. It is not clear how s/he is supervising this research. This should be illustrated and clarified.
  6. It is not clear whether the research participates are recruited randomly in your research.
  7. Significance of the research and statement of the problem should be addressed and elaborated.

Reviewer 3 Report

  • The format of the paper is wrong.
  • Please avoid all “I” and “We” statements on the paper.
  • Please edit the grammar as the paper contains a lot of mistakes.
  • For the introduction chapter, the chapter structured was too random.
  1. Please add a section of purpose
  2. Please add a section of significance.
  3. There is no connection why this research is significant and important to the community.
  4. The focus of this chapter was failed.
  • For the methodology chapter.
  1. What these groups of participants?
  2. The inclusion and exclusion criteria of the participants.
  3. How can you recruit these groups of people?
  4. What about human subject protection?
  • The result chapter,
  1. The structure of the chapter was not professional. Very hard to follow.
  2. There are no connection and the statements were not clear.
  • The discussion chapter,
  1. It was written unprofessionally.
  2. I cannot see how the author can connected the previous studies into this paper.
  • Conclusion
  1. The authors should add a limitation section.
  2. The authors should add a future research direction section.
  3. The authors should add an implementation section.
  4. The conclusion does not really conclude anything.

Reviewer 4 Report

Dear authors, below you will find my comments on your work. I hope they are helpful to improve your work.

1.-The abstract has errors in English writing. Overall, review the whole document.

2.-What are the characteristics of the software used for data analysis?

3.-It is suggested to discuss, as far as possible, if the findings are due to the socio-cultural characteristics of the area where the study was applied, and if appropriate indicate what these characteristics could be.

4.-Table 1 is not referenced or commented on.

5.-It is suggested to summarize the demographic data of the participants in a table.

6.-It is suggested to include other intervention strategies to address the problem, for example educating the population, especially regarding the dignified treatment of people with this condition.